# Structure-based identification of bioactive compounds as trace amine-associated receptor 1 agonists for the therapeutic management of major depressive disorder

Abdelbaset Mohamed Elasbali[1,2], Ahmed S. Ali[2,3], Mohd Adnan[4], Taj Mohammad[5], Anas Shamsi[6]*, Md. Imtaiyaz Hassan[5]*

**1** Department of Clinical Laboratory Science, College of Applied Medical Sciences-Qurayyat, Jouf University, Sakakah, Saudi Arabia, **2** King Salman Center for Disability Research, Riyadh 11614, Saudi Arabia, **3** Department of Physical Therapy and Health Rehabilitation, College of Applied Medical Sciences - Qurayyat, Jouf University, Saudi Arabia, **4** Department of Biology, College of Science, University of Ha'il, Ha'il, Saudi Arabia, **5** Centre for Interdisciplinary Research in Basic Sciences, Jamia Millia Islamia, New Delhi, India, **6** Centre of Medical and Bio-Allied Health Sciences Research, Ajman University, Ajman, United Arab Emirates

\* anas.shamsi18@gmail.com (AS); mihassan@jmi.ac.in (MIH)

## Abstract

The global burden of major depressive disorder (MDD) drives ongoing efforts to develop safer and more targeted treatment strategies. Modern advances have identified trace amine-associated receptor 1 (TAAR1) as a promising non-monoaminergic target with demonstrated efficacy in treating neuropsychiatric conditions, including MDD. Discovering TAAR1 agonists holds promise for modulating neuropsychiatric disorders while potentially reducing the common side effects associated with conventional therapies. This study employed a structure-based virtual screening approach to identify potential TAAR1 agonists from the IMPPAT database, a curated collection of Indian medicinal plant-derived bioactive phytoconstituents. The initial filtering was done on the compounds based on Lipinski's rule of five, which was followed by molecular docking, PAINS screening, pharmacokinetic evaluation, and bioactivity predictions. Through this integrative screening approach, we discovered two promising phytochemicals, Bianthraquinone and Peimisine, demonstrating strong binding affinities and favorable drug-like properties. Detailed interaction analysis revealed that both compounds formed stable hydrogen bonds, hydrophobic contacts, and π-π stacking interactions with key residues within the TAAR1 binding pocket, contributing to their high binding stability and receptor specificity. All-atom molecular dynamics simulations, MM-PBSA, and essential dynamics analyses affirmed that they were stable and exhibited favorable conformational interactions. These findings highlight the therapeutic potential of naturally derived TAAR1 agonists and support their further exploration as next-generation antidepressants, laying the foundation for future experimental and clinical development.

**Data availability statement:** All relevant data are within the manuscript and its Supporting Information files.

**Funding:** The authors extend their appreciation to the King Salman Center for Disability Research for funding this work through Research Group Number KSRG-2024-285.

**Competing interests:** The authors have declared that no competing interests exist.

## 1. Introduction

Major depressive disorder (MDD) is one of the most prevalent and debilitating mental health conditions, affecting over 280 million individuals globally [1]. It is characterized by persistent low mood, anhedonia, cognitive impairment, and, in severe cases, suicidal ideation [2]. The multifaceted impact of MDD extends beyond emotional distress to include metabolic dysfunction, neurocognitive decline, and impaired quality of life, making it a significant public health concern [3]. Despite numerous decades of research, the therapeutic landscape remains suboptimal, and many patients do not respond adequately to first-line antidepressants or suffer from intolerable side effects [4]. Initial development of traditional antidepressants was based on the monoamine hypothesis, which attributes depressive symptoms to deficiencies in serotonin (5-HT), norepinephrine (NE), and dopamine (DA) [5]. Despite their efficacy, these treatments have a long time to onset of action, small response rates, and undesirable side effects, highlighting the need for novel mechanistic targets and more tolerable therapeutic options [6].

New evidence assigns the trace amine-associated receptor 1 (TAAR1) to the ranks of promising pharmacological targets in neuropsychiatry [7]. TAAR1 is a G protein-coupled receptor (GPCR) that indirectly modulates monoaminergic neurotransmission by affecting dopaminergic, serotonergic, and noradrenergic tone [8]. Unlike conventional antidepressants, TAAR1 agonists do not directly antagonize dopamine receptors and therefore avoid extrapyramidal symptoms and metabolic side effects [9]. TAAR1 activation has been shown to normalize dopaminergic signaling and to improve cognitive function, as well as to produce antidepressant-like effects in animal models of MDD in preclinical studies [9,10]. Thus, this target was further validated in patients with schizophrenia and MDD by clinical trials with dual TAAR1 and 5-HT1A receptor agonist Ulotaront (SEP-363856), which demonstrated efficacy and a favorable safety profile [11,12].

Despite the therapeutic promise of TAAR1, identifying safe and efficacious agonists with adequate brain bioavailability remains a challenge [13]. Most current efforts focus on synthetic compounds, which often require extensive optimization and may present unforeseen pharmacokinetic or toxicity issues [14]. This creates a compelling need to explore alternative sources of TAAR1 agonists, particularly those with a proven track record of biological activity and favorable safety profiles. Natural products represent a valuable chemical space for central nervous system (CNS) drug discovery, owing to their structural diversity, generally lower toxicity, and longstanding use in traditional medicine [15]. In particular, phytochemicals derived from medicinal plants have historically played a pivotal role in the identification and development of therapeutic agents due to their chemical complexity, target promiscuity, and favorable safety profiles [16]. Among these, compounds capable of crossing the blood–brain barrier and modulating multiple neural targets are especially promising, positioning them as ideal candidates for TAAR1 modulation in the treatment of neuropsychiatric disorders.

Exploring phytochemicals has now entered a rational and targeted phase, driven by advances in high-throughput computational technologies [17]. In silico screening methods provide the fast evaluation of vast phytochemical libraries against a specific protein target with good precision in predicting binding affinity and key molecular interactions [18]. Such pharmacokinetic and toxicity profiling helps identify compounds with favorable drug likeness, safety, and therapeutic potential [19]. Additionally, molecular dynamics (MD) simulations, essential dynamics, and molecular mechanics-Poisson-Boltzmann surface area (MM-PBSA) analysis give crucial information about ligand-receptor complexes in solvent conditions of the stability, conformational flexibility, and dynamic behavior, contributing to a better understanding of the mechanisms of receptor activation [20].

This study used a multi-level computational framework to analyze the possibility of phytochemicals as TAAR1 agonists for the therapeutic development of MDD. Our approach integrates structure-based virtual screening, pharmacokinetic evaluation, and advanced molecular simulation techniques to discover natural compounds with high affinity for TAAR1, robust pharmacological profiles, and potential therapeutic efficacy. In addition to helping to identify novel lead compounds, this strategy also fits within the higher-level goal of precision psychiatry, which is to provide safer, more targeted, and more effective treatment for individuals suffering from MDD.

## 2. Materials and methods

### 2.1. Input data and computational resources

To identify potential TAAR1 agonists, we utilized a combination of bioinformatics tools and publicly available databases in a virtual screening workflow. Molecular docking simulations were conducted using InstaDock v1.2 [21] to evaluate various phytoconstituents' binding affinity and pose within the TAAR1 binding pocket. The 3D structure of the TAAR1 receptor (PDB ID: 9JKQ, resolution: 2.66 Å) was obtained from the RCSB Protein Data Bank [22]. A curated library of small-molecule phytochemicals was sourced from IMPPAT 2.0 (https://cb.imsc.res.in/imppat/), which hosts comprehensive information on plant-derived bioactive compounds [23]. To ensure drug-likeness, all compounds in the IMPPAT library were initially screened using Lipinski's rule of five (Ro5), which considers key physicochemical parameters including molecular weight (≤ 500 Da), lipophilicity (LogP ≤ 5), hydrogen bond donors (≤ 5), and hydrogen bond acceptors (≤ 10). Only compounds meeting these criteria were selected for subsequent molecular docking analyses. After the docking screening, the docked protein–ligand complexes were further analyzed and visualized in 3D and 2D formats using PyMOL [24] and Discovery Studio Visualizer [25], facilitating detailed examination of binding interactions. Deep-PK (https://biosig.lab.uq.edu.au/deeppk/) [26] was employed to predict their ADMET (absorption, distribution, metabolism, excretion, and toxicity) profiles to evaluate the pharmacokinetic behavior of the screened compounds. Furthermore, the PASS online server (https://www.way2drug.com/passonline/) [26] was used to assess the probable biological activities of the selected compounds.

### 2.2. Molecular docking-based screening

The most common use of computer-aided drug discovery is molecular docking, which predicts the preferred binding position of small molecules in the active site of a receptor with high affinity. Here, we used molecular docking to investigate the interaction between molecules from the IMPPAT library of phytochemicals with TAAR1. We used InstaDock v1.2 software (https://www.hassanlab.in/instadock) for molecular docking to execute fast and reliable predictions. InstaDock was chosen for this study due to its simplified, user-friendly GUI and its capability for high-throughput virtual screening, especially suited for large phytochemical libraries. The grid box dimensions for TAAR1 were set to 20 Å × 26 Å × 20 Å around X: −1.182 Å, Y: −4.706 Å, and Z: 8.778 Å, defined on X, Y, and Z coordinates, respectively. The spacing was kept at 1 Å, and other docking parameters were applied at default settings of InstaDock. These coordinates were set based on the binding profile of the reference co-crystallized agonist of TAAR1, Ulotaront [27]. To validate the reliability of the docking protocol, a retrospective redocking analysis was performed. In this procedure, the co-crystallized ligand Ulotaront was redocked into the TAAR1 binding pocket of

the crystal structure (PDB ID: 8JLO). The predicted binding pose closely aligned with the experimentally determined crystallographic conformation, as evidenced by a low root-mean-square deviation (RMSD) of 0.198 Å (S1 Fig). This high degree of overlap confirms the accuracy and robustness of the docking protocol in reproducing the native ligand orientation within the TAAR1 active site. The selection of potential lead compounds was made based on binding affinity scores. Docked poses with the lowest scores were analyzed to depict the type of interaction using PyMOL [24] and Discovery Studio Visualizer [25]. These powerful tools facilitated comprehensive representation of ligand-receptor interactions, allowing for assessment of significant non-bonded interactions, including hydrogen bond donor-acceptor pairings, hydrophobic interactions, solvation effects, and ionic attractions, which stabilize interactions within the bound conformation. The top-performing compounds were selected for further pharmacokinetic and dynamic stability testing.

## 2.3. ADMET analyses

Early-stage drug discovery necessitates determinations of ADMET properties of drug candidates for their pharmacokinetic viability and safety. We used the Deep-PK web-based tool [28] to assess the ADMET features of the filtered phytochemicals. Deep-PK tool uses molecular descriptors and machine learning algorithms to determine drug-likeness, bioavailability, and potential toxicity risks. Screened compounds were evaluated for key pharmacokinetic properties such as GI absorption, BBB permeability, cytochrome P450 interactions, renal clearance, and other potential toxicity liabilities (e.g., hepatotoxicity, mutagenicity). Here, we filtered out from the dataset molecules that displayed strong, non-specific interactions using the Pan-Assay Interference Compounds (PAINS) filter [29]. Based on our ADMET analysis, we could filter out the compounds that did not satisfy the required parameters due to bioavailability issues, high toxicity, or poor metabolic stability, leaving only drug-like molecules for further investigation.

## 2.4. Biological activity prediction

To analyze the therapeutic potential of the screened compounds, we predicted biological activities using the online PASS (Prediction of Activity Spectra for Substances) server [26]. The tool uses a compound's structural similarity to known bioactive molecules to give a probability version of how it would act on a given target. The PASS server calculates Pa (probability of active) and Pi (probability of inactive). This preliminary step enabled us to focus on molecules with higher probabilities of having antipsychotic/neuroprotective effects, matching the goal of this study of identifying TAAR1-targeted phytochemicals.

## 2.5. Interaction analysis

Interaction analysis is an important tool to better understand the binding affinity and selectivity of a ligand to its target protein [30]. The analysis of interactions between proteins and ligands gives information about the stability of the binding and the structure-activity relationship, which contributes to practicality for lead optimization. Previous studies report that the TAAR1 binding pocket includes key residues such as Asp103, Ser107, Trp264, Phe268, Tyr294, and Cys182 [31,32]. These residues facilitate critical interactions, including hydrogen bonding, π-π stacking, and hydrophobic contacts essential for ligand binding and receptor activation. We analyzed the binding conformations of the selected compounds inside the TAAR1 binding pocket using PyMOL [24] and Discovery Studio Visualizer [25] to examine the interaction profiles of the selected compounds. Key molecular interactions, such as hydrogen bonds, hydrophobic contacts, π–π stacking interactions, and salt bridges, played an important role to stabilize the ligand within the binding site and promoting receptor activation. Ligand orientation with strong, specific interactions to critical active site residues of TAAR1 was a major criterion in the selection process. The ligands showing good binding energies and substantial molecular interactions were selected for subsequent molecular dynamics studies to assess their stability in physiologically relevant environments.

 

## 2.6. Molecular dynamics simulations

Molecular dynamics (MD) simulations offer a unique window into the atomic motion and dynamic behavior of biomolecular complexes for understanding the timescale and persistence of protein–ligand interactions [20]. In this study, 500 ns MD simulations were conducted to evaluate the stability, conformational flexibility, and interaction dynamics of TAAR1 in complex with the identified phytochemicals. The simulations were performed using GROMACS v5.5.1, a popular molecular dynamics package used in simulating biomolecular systems [33], which has been widely used owing to its efficiency. We used the PRODRG server (https://prodrg1.dyndns.org/) [34] to build molecular topology files for the ligand molecules. This led us to prepare the protein-ligand complex systems of TAAR1 with the elucidated compounds. PRODRG has known limitations, including reduced accuracy in generating torsional parameters and the absence of sophisticated charge models, which can affect the accuracy of ligand topology in MD simulations. All simulations used the GROMOS 54A7 force field [35] for modelling atomic interactions. A cubic simulation box with a minimum distance of 1 nm was used to solvate each system. A simple point charge (SPC216) [36] water model was employed to represent physiologic conditions. In this stage, counter ions were included to screen the overall charge of the system. An energy minimization step (using 1500 iterations of the steepest descent algorithm) was applied to eliminate any steric clashes between all atom selection and to achieve a minimized and energetically stable structure. After minimization, equilibration was conducted using an NVT (constant number of particles, volume, and temperature) followed by an NPT (constant number of particles, pressure, and temperature) ensemble. Bond lengths were constrained using the LINCS algorithm [37], and water molecules were stabilized using SETTLE [38]. Temperature control was performed using a modified Berendsen thermostat [39] at 300 K. The Berendsen Barostat stabilized pressure at 1 bar. Using the Leapfrog method, a time step of 2 fs was used to integrate the equations of motion. Each system's production stage of the simulation was performed for 500 ns to obtain trajectory data for the analysis. Various GROMACS (gmx modules) tools were used to assess structural deviations, fluctuations, compactness, and interaction stability of the systems.

## 2.7. Principal component analysis

Principal component analysis (PCA) is a crucial computational tool for investigating the conformational flexibility and thermodynamics of protein-ligand systems [40]. These reveal the leading motions and the most stable conformations taken by biomolecular systems. PCA was conducted to analyze the predominant motions of TAAR1 in its unbound and ligand-bound forms. The covariance matrix of atomic movements was calculated over the Cα atoms of the protein. Then, it was subjected to eigenvector decomposition to extract the principal components (PCs) that contribute to the global motion. Results for the first few eigenvectors that account for the most excellent conformational flexibility were used to study the dynamic behavior of unbound and ligand-bound TAAR1. An increase in structural rigidity would be evident from decreased large-scale motions upon ligand engagement.

## 2.8. Free energy landscapes

We performed free energy landscape (FEL) analysis [40] to characterize the energy landscape of TAAR1 and its complexes with the selected compounds. We generated the FEL using this distribution's first two principal components (PC1 and PC2) as reaction coordinates, offering an overview of the more thermodynamically representative conformations. Lower energy regions on the FEL correspond to stable structural states, while higher energy regions indicate less favorable or transient conformations. PCA and FEL analysis have also successfully integrated to reveal key insights into the stability of the TAAR1 receptor and its folding mechanisms upon ligand binding. This analysis further corroborated the findings and supported the conclusion that the identified phytochemicals stabilise TAAR1, positioning them as potential leads for TAAR1 agonists. However, PCA and FEL approaches rely on harmonic approximations of motion and may fail to capture rare conformational transitions, which represent a potential limitation.

## 2.9. MM-PBSA calculations

The binding free energy between the protein and ligand was estimated using the Molecular Mechanics Poisson–Boltzmann Surface Area (MM-PBSA) method [41]. Calculations were performed using the *g_mmpbsa* tool, an open-source package integrated with GROMACS [42]. A 10-ns stable segment of the MD trajectory, saved at 10-ps intervals, was used for energy estimations. The binding free energy ($\Delta G_{binding}$) was computed as:

$$\Delta G_{binding} = G_{complex} - (G_{receptor} + G_{ligand})$$

where $G_{binding}$, $G_{receptor}$, and $G_{ligand}$ represent the free energies of the complex, receptor, and ligand, respectively. The tool also provides per-residue energy contributions through post-processing scripts included in the package.

## 3. Results and discussion

### 3.1. Molecular docking screening

Molecular docking is a crucial computational method in modern drug discovery that helps identify potential candidates by predicting their binding affinity and stability of the interaction with target proteins [43]. Molecular docking was used in this study to screen a library of phytochemicals as potential TAAR1 agonists. To streamline the selection process, compounds from the IMPPAT-2 library (~18,000 compounds) were screened according to Lipinski's rule of five, characterized by drug-likeness using molecular weight, lipophilicity, hydrogen bond donors and acceptors, and by molar refractivity. This initial filtration yielded a dataset of 11,908 phytochemicals further analyzed through molecular docking on InstaDock. The binding affinities of these compounds with TAAR1 were calculated, where the top 10 candidates were selected based on their docking score and ligand efficiency. The binding free energy of these top-scored compounds was found to be in the range of −9.3 to −10.2 kcal/mol, suggesting strong binding interactions with TAAR1 (Table 1). The predicted lower binding energy indicated a more stable and favorable ligand-receptor interaction, consistent with the assumption that these compounds can act as potent TAAR1 agonists. Ligand efficiency, in addition to docking scores, was used as a key parameter for compound ranking [44].

All the chosen hits also had ligand efficiency values greater than 0.25 kcal/mol/non-H atom, suggesting a capacity for favorable binding and an appropriate molecular size. Ligand efficiency is an important property in drug design, ensuring that the compound does not create many additional atoms to reach the required binding [44]. Based on the molecular

**Table 1. The top-ranked phytochemicals identified through molecular docking against TAAR1\*.**

| S. No. | Phytochemical ID | Phytochemical Name | Source | Binding Affinity (kcal/mol) | Ligand Efficiency |
|--------|------------------|--------------------|--------|----------------------------|-------------------|
| 1. | IMPHY014146 | Bianthraquinone | *Senna hirsuta* | −10.2 | 0.3187 |
| 2. | IMPHY014795 | Absinthin | *Artemisia absinthium* | −10.0 | 0.2778 |
| 3. | IMPHY001309 | Anabsinthin | *Artemisia absinthium* | −9.9 | 0.275 |
| 4. | IMPHY001734 | Peimisine | *Fritillaria cirrhosa* | −9.5 | 0.3065 |
| 5. | IMPHY012553 | Millettone | *Piscidia piscipula* | −9.4 | 0.3357 |
| 6. | IMPHY000058 | Ovalichromene B | *Pongamia pinnata* | −9.3 | 0.3577 |
| 7. | IMPHY000366 | Jervine | *Veratrum viride* | −9.3 | 0.3 |
| 8. | IMPHY005775 | Gentrogenin | *Agave sisalana* | −9.3 | 0.3 |
| 9. | IMPHY006510 | Quecellin | *Citrullus colocynthis* | −9.3 | 0.2735 |
| 10. | IMPHY008834 | Withaphysalin C | *Physalis indica* | −9.3 | 0.2657 |
| 11. | Ulotaront | | | −4.7 | 0.3917 |

\*Table lists binding free energy values (in kcal/mol) and ligand efficiency (kcal/mol/non-H atom).

docking results, these top 10 phytochemicals showed strong interactions with TAAR1 compared to the reference agonist, Ulotaront, which showed an affinity of −4.7 kcal/mol. These results serve as a solid base for further exploration and validation of these compounds for their drug-likeness.

## 3.2. ADMET properties

Assessment of ADMET properties is the backbone of the modern drug discovery process, on which pharmacokinetics of drug candidates and their safety can be analyzed [45]. ADMET profiling of a chemical compound is a way to ensure drug-likeness by evaluating its key properties like solubility, permeability, metabolic stability, and toxicity risk. We used the Deep-PK web server to predict the ADMET properties of the top 10 compounds selected via molecular docking. This web server is a robust computational tool that allows users to predict pharmacokinetic and toxicity properties from the molecular structure [28]. Different ADMET parameters were revealed, such as GI absorption, BBB permeability, cytochrome P450 interactions, renal clearance, and hepatotoxicity risk (Table 2). The collection was further refined using the PAINS filter to remove compounds that showed undesirable bioactivity, leading to false dimensionality in experiments. Using these criteria, two compounds, Bianthraquinone and Peimisine, out of the ten compounds were selected, as they exhibited optimal pharmacokinetic properties, high bioavailability, and no toxicity risks. Notably, they also exhibited BBB permeability, a critical requirement for treating neurological disorders such as MDD, akin to the reference compound [46]. Given their acceptable metabolic stability and lack of toxicological concerns, these phytochemicals represent promising lead candidates for further optimization and experimental validation in the context of TAAR1-targeted drug discovery.

## 3.3. PASS analysis

Predicting biological activity is an essential stage of computational drug development, enabling researchers to evaluate the possibility of therapeutic effects for candidate compounds [47]. The PASS server is one of the most popular computational tools that predicts the biological activity of small molecules based on structure-activity relationship (SAR) models, based on learning from a large clinical and preclinical dataset [26]. The compounds that passed the ADMET screening above were further analyzed using a PASS to identify their probable pharmacological activities. The prediction starts with two probabilities: Pa (active) and Pi (inactive). The results showed that both compounds, Bianthraquinone and Peimisine,

**Table 2. The predicted ADMET properties of the selected phytochemicals targeting TAAR1\*.**

| S. No. | Phytochemical | Absorption (Oral Bioavailability) | Distribution (BBB permeability) | Metabolism (CYP 3A4 Inhibitor) | Excretion (OCT2 Inhibitor) | Toxicity (AMES & hERG) | Elimination criterion |
|---|---|---|---|---|---|---|---|
| 1. | Bianthraquinone | Bioavailable | Penetrable | No | No | Safe | None |
| 2. | Absinthin | Bioavailable | No | No | No | Toxic | DT |
| 3. | Anabsinthin | Non-Bioavailable | No | No | No | Toxic | ADT |
| 4. | Peimisine | Bioavailable | Penetrable | No | No | Safe | None |
| 5. | Millettone | Non-Bioavailable | Penetrable | Yes | No | Toxic | AMT |
| 6. | Ovalichromene B | Bioavailable | Penetrable | Yes | No | Toxic | MT |
| 7. | Jervine | Bioavailable | Penetrable | No | Yes | Toxic | ET |
| 8. | Gentrogenin | Bioavailable | Penetrable | No | Yes | Toxic | ET |
| 9. | Quecellin | Bioavailable | No | No | No | Toxic | DT |
| 10. | Withaphysalin C | Bioavailable | Penetrable | No | No | Toxic | T |
| 11. | Ulotaront | Bioavailable | Penetrable | No | No | Safe | None |

\*The table summarizes key pharmacokinetic parameters such as gastrointestinal absorption, blood-brain barrier permeability, metabolic stability, and toxicity risks. HIA, Human intestinal absorption; BBB, the blood-brain barrier.

had a high probability of exhibiting antineurotic, antipsychotic, and neuroprotective activities, offering further potential for targeting TAAR1 in the treatment of MDD (Table 3). They demonstrate relevant indicators of biological activities, which strengthens the rationale for their further development as antidepressant drugs.

### 3.4. Interaction analysis

To analyze the binding potential of Bianthraquinone and Peimisine with TAAR1, an in-depth protein-ligand interaction analysis was conducted to observe their interactions. The docking conformations of these compounds were studied in PyMOL and Discovery Studio Visualizer to evaluate their molecular interactions (Fig 1). Both Bianthraquinone and Peimisine, along with the reference compound Ulotaront, displayed prominent binding to the TAAR1 binding site and made multiple contacts with important catalytic and regulatory residues (Fig 1A). Of the interacting amino acids, the ligand binding site Asp103 formed significant interactions with both molecules (Fig 1B–C). Such interactions imply that these selected phytochemicals could modulate TAAR1 and act as agonists to regulate TAAR1 (Fig 1D). The structural modeling confirmed that both compounds indeed fit within the TAAR1 binding pocket and established an array of hydrogen bonds, hydrophobic, and electrostatic interactions. These compounds exhibit robust binding interactions at the receptor site, indicating their potential as efficient TAAR1 agonists. These results support the hypothesis that Bianthraquinone and Peimisine represent promising lead molecules for development as TAAR1-targeted therapeutics.

Detailed analysis of non-covalent interactions of Bianthraquinone and Peimisine with TAAR1 provided more profound insights into their binding mechanisms. Knowing the detailed molecular interactions related to ligand binding is the key to understanding the receptor's modulation and optimizing drug candidates. The interaction modes of Bianthraquinone and Peimisine were analysed in detail with Discovery Studio Visualizer for their stabilising forces by hydrogen bonding, hydrophobic interactions, and van der Waals forces (Fig 2). The 2D interaction plots yielded details of these molecular contacts, with a broad characterisation. TAAR1-Bianthraquinone complex formed four hydrogen bonds with Arg83, Ser107, Ser108, and Tyr294, one sigma bond with Ile290, one pi-sulfur bond with Cy182, pi-anion bond with Asp103, two alkyl bonds with Thr100 and Val184, two Pi-pi shaped bonds with Trp264 and Phe268, and ten van der Waals interactions

**Table 3. Biological activity predictions of the selected phytochemicals based on PASS analysis\*.**

| S. No. | Compound | $P_a$ | $P_i$ | Biological Activity |
|---|---|---|---|---|
| 1. | Bianthraquinone | 0,849 | 0,017 | Phobic disorders treatment |
|  |  | 0,767 | 0,005 | Neurotransmitter uptake inhibitor |
|  |  | 0,636 | 0,060 | Antineurotic |
|  |  | 0,557 | 0,022 | Neurotransmitter antagonist |
|  |  | 0,443 | 0,114 | Acute neurologic disorders treatment |
| 2. | Peimisine | 0,465 | 0,097 | Acetylcholine neuromuscular blocking agent |
|  |  | 0,485 | 0,019 | Menopausal disorders treatment |
|  |  | 0,489 | 0,043 | Antinociceptive |
|  |  | 0,372 | 0,070 | Dementia treatment |
|  |  | 0,284 | 0,043 | Vascular dementia treatment |
| 3. | Ulotaront | 0,402 | 0,142 | Acute neurologic disorders treatment |
|  |  | 0,314 | 0,193 | Neurotransmitter uptake inhibitor |
|  |  | 0,332 | 0,215 | Antineurotic |
|  |  | 0,303 | 0,051 | Antipsychotic |
|  |  | 0,038 | 0,033 | Neuropathy treatment |

\*The table presents the probability of activity (Pa) and probability of inactivity (Pi) values, indicating the potential pharmacological relevance of each compound in TAAR1 agonism.

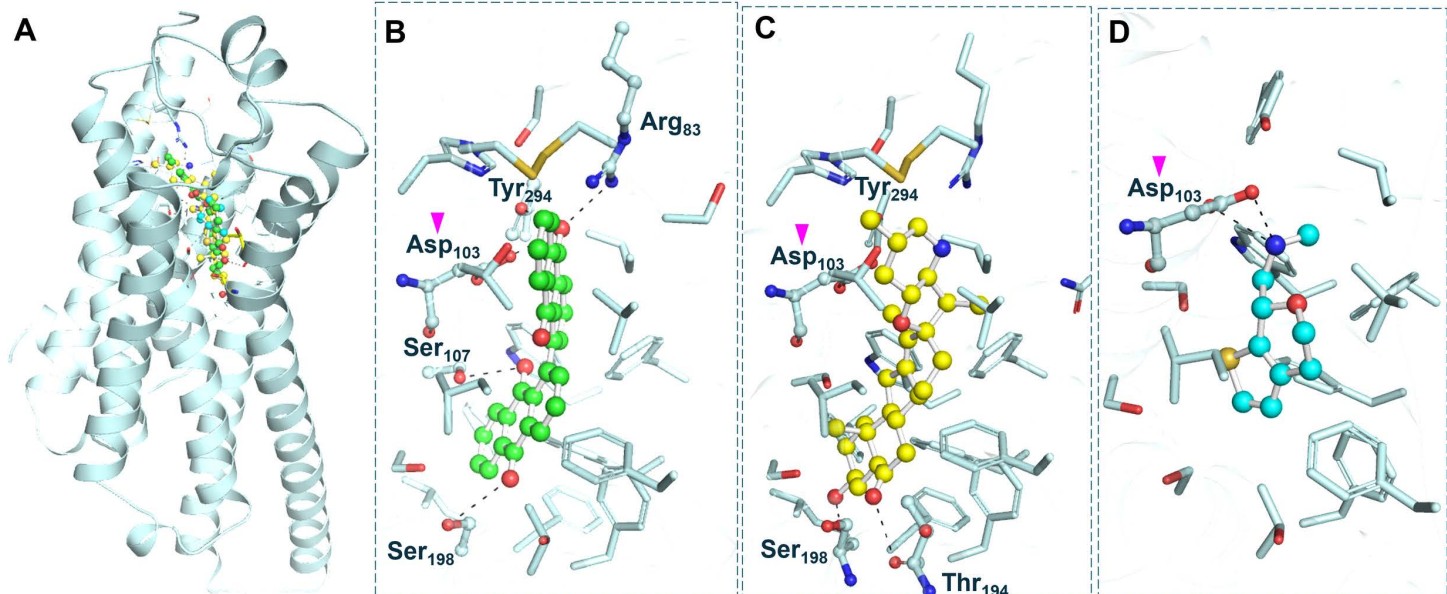

**Fig 1. Structural representation of TAAR1 in complex with the selected phytochemicals, Bianthraquinone (green) and Peimisine (yellow), and the reference TAAR1 agonist Ulotaront (cyan).** (A) Cartoon representation of the TAAR1 structure, with the bound compounds shown occupying the TAAR1 binding pocket. (B) A zoomed-in view of TAAR1, illustrating the precise positioning of Bianthraquinone (C), Peimisine, and (D) Ulotaront. (C) The amino acid residues forming hydrogen bonds are labelled. The magenta arrow highlights the TAAR1 binding site, Asp103. The figure was generated through PyMOL using the structural coordinates from the docking study.

with various residues (Fig 2A). At the same time, TAAR1-Peimisine complex formed seven carbon-hydrogen bonds with His99, Asp103, Ser107, Ile111, Val184, Ser198, and Tyr268, three alkyl bonds with Ile104, Trp264, and Phe268, and twelve van der Waals interactions with various residues (Fig 2B). Both compounds showed similar interactions with the reference TAAR1 agonist, Ulotaront (Fig 2C). Thestrong binding affinity and stability of compounds in the receptor pocket are attributed to these interactions. Both Bianthraquinone and Peimisine exhibited strong interactions with the binding site residue Asp103, indicating a conserved binding pattern for both compounds. A detailed overview of all major non-covalent interactions, including hydrogen bonding, π–π stacking, and hydrophobic contacts, provides a clearer view of structure-function relationships (Table 4).

These interactions suggest the ability of the compounds to allosterically modulate TAAR1 activity by stabilizing important receptor conformational states. Structure-activity relationship (SAR) analysis indicates that the reference TAAR1 agonist contains hydrophobic aromatic rings that facilitate π–π interactions with Trp264 and Phe268, and polar functional groups that form hydrogen bonds with Asp103 and Ser107 [31]. The analysis indicates that a basic nitrogen near Asp103 seems crucial for ionic and hydrogen bond interactions. Both Bianthraquinone and Peimisine exhibit these features, suggesting potential agonism. Scaffold optimization could involve introducing substituents to enhance π–π stacking or polarity around the Asp103 region to improve binding affinity and selectivity. So, considering the binding interactions of these compounds with TAAR1, we performed all-atom MD simulations to check their stability and dynamic behavior.

### 3.5. MD simulation analysis

Protein structural dynamics may change after they interact with the ligands [48]. Studying these dynamics can give insight into the binding mechanism for and stability of small molecules to a given protein target. Here, all-atom MD simulations were performed to see the behavior of TAAR1 in the presence of Bianthraquinone and Peimisine. Using periodic boundary

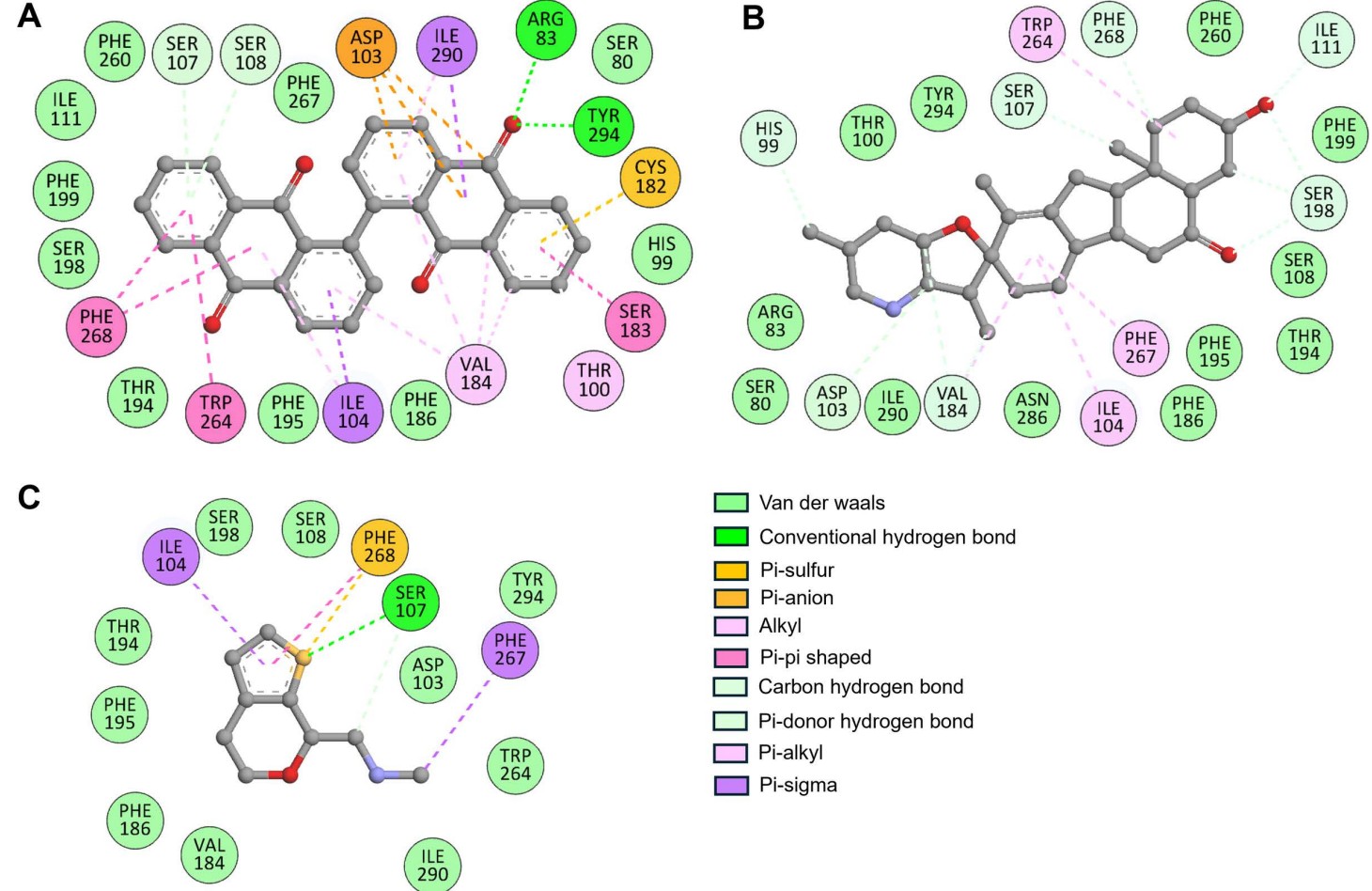

**Fig 2. Two-dimensional depiction of ligand-receptor interactions between TAAR1 and the selected phytochemicals.** (A) The interaction map of TAAR1 with Bianthraquinone shows hydrogen bonding and other non-covalent interactions. (B) The interaction map of TAAR1 with Peimisine illustrates similar binding patterns and interactions with binding site residues. (C) The interaction map of TAAR1 with Ulotaront shows hydrogen bonding and other non-covalent interactions.

conditions, four systems, TAAR1, TAAR1-Bianthraquinone, TAAR1-Peimisine, and TAAR1-Ulotaront, were simulated for 500 ns. The aim was to monitor and analyze the time evolution of different structural features and to obtain an insight into the dynamics of TAAR1 before and after binding, with Bianthraquinone and Peimisine.

**3.5.1. Structural dynamics.** The overall stability of the protein and its complexes was assessed using RMSD analysis [49]. The trajectories showed that all the ligand-bound systems were stable with a few RMSD fluctuations (Fig 3A). While the TAAR1-Bianthraquinone complex experienced the least deviation and remained relatively stable throughout the simulation, indicating that Bianthraquinone forms a strong and stable interaction with TAAR1. The profile for the TAAR1-Peimisine complex was also relatively stable, albeit with slight fluctuations more significant than those of TAAR1-Bianthraquinone, indicating small conformational changes upon binding. The probability density function (PDF) of RMSD values confirmed this view as the ligand-bound systems retained a more compact structural arrangement than the unbound receptor (Fig 3A, lower panel). These results suggest that ligand binding enhances structural compactness and contributes to stabilizing functionally relevant conformational states. This dynamic stabilization may underlie the potential agonistic activity of Bianthraquinone and Peimisine at the TAAR1 receptor.

**Table 4. Interacting residues and their types within TAAR1–ligand complexes.** The table summarizes key amino acid residues involved in hydrogen bonding, π–π stacking, π–alkyl, π–sulfur, electrostatic, and van der Waals interactions with Bianthraquinone, Peimisine, and Ulotaront.

| Ligand | Interaction Type | Participating Residues |
|---|---|---|
| Bianthra-quinone | Hydrogen bond | Arg83, Ser107, Ser108, Tyr294 |
| | π–π interaction | Ser183, Trp264, Phe268 |
| | π–alkyl | Thr100, Val184 |
| | π-anion | Asp103 |
| | π-sigma | Ile104 |
| | Sulfur interaction | Cys182 |
| | Van der Waals | Ser80, His99, Ile111, Phe186, Thr194, Phe195, Ser198, Phe199, Phe260, Phe267 |
| Peimisine | Hydrogen bond (Carbon hydrogen/π-donor) | His99, Asp103, Ser107, Ile111, Val184, Ser198 |
| | π–π interaction | Phe267, Trp264 |
| | π–alkyl | Ile104 |
| | Van der Waals | Ser80, Arg83, Thr100, Ser108, Phe186, Thr194, Phe195, Ser198, Phe199, Phe260, Asn286, Ile290, Tyr294 |
| Ulotaront | Hydrogen bond | Ser107 |
| | π-sulfur | Phe268 |
| | π–sigma | Ile104, Phe267 |
| | Van der Waals | Asp103, Ser108, Val184, Phe186, Thr194, Phe195, Ser198, Trp264, Ile290, Tyr294 |

We conducted the root-mean-square-fluctuation (RMSF) analysis to better understand how ligand binding influences the stability of TAAR1's structure at the residual level. This enabled the flexibility of individual residues of the protein to be examined. RMSF analysis showed almost similar fluctuation patterns in all the systems (Fig 3B). Bianthraquinone and Ulotaront reduce fluctuations, especially in binding hot spots, emphasizing their stabilizing role on the receptor. The TAAR1-Bianthraquinone complex showed the least variability among several important structural features, suggesting a stronger and more rigid interaction. However, the TAAR1-Peimisine interfacial regions demonstrated considerably higher fluctuations that potentially indicate fine-tuning of the structure around the binding cavity in response to ligand binding. The results demonstrate that ligand binding dynamically stabilizes TAAR1, and both Bianthraquinone and Peimisine contribute notably to this enhanced conformational stability. A decreased flexibility in ligand-bound arrangements indicates that these phytochemicals can assist TAAR1 in maintaining a proper conformation toward folding; it can be essential before functional agonism. Because receptor stability is vital for drug efficacy, the observation that these compounds have the potential to reduce structural fluctuations suggests they are viable options for developing TAAR1-targeting agents.

**3.5.2. Structural compactness.** The compactness of a protein structure serves the critical aspect of stabilizing protein function and Integrity [50]. We explored the folding characteristics of TAAR1 in both free and ligand-bound forms, using radius of gyration ($R_g$) and solvent-accessible surface area (SASA) as crucial parameters to assess the compactness (or folding) of the protein. To determine if ligand binding affected the global folding and compactness of TAAR1, the $R_g$ was analyzed from the simulated trajectories (Fig 4A). A lower $R_g$ value means a more compact structure, and a higher $R_g$ value means a more expanded conformation. For TAAR1, we found average $R_g$ values during each simulation to be similar across all systems (TAAR1, TAAR1-Bianthraquinone, TAAR1-Peimisine, and TAAR1-Ulotaront), ranging from 2.1 nm to 2.3 nm (Table 5). This indicates that ligand binding did not cause a pronounced conformational change to the receptor, preserving the structural integrity of the protein. The distance distribution of $R_g$ values further confirmed the stabilizing role of Bianthraquinone and Peimisine on TAAR1. The PDF

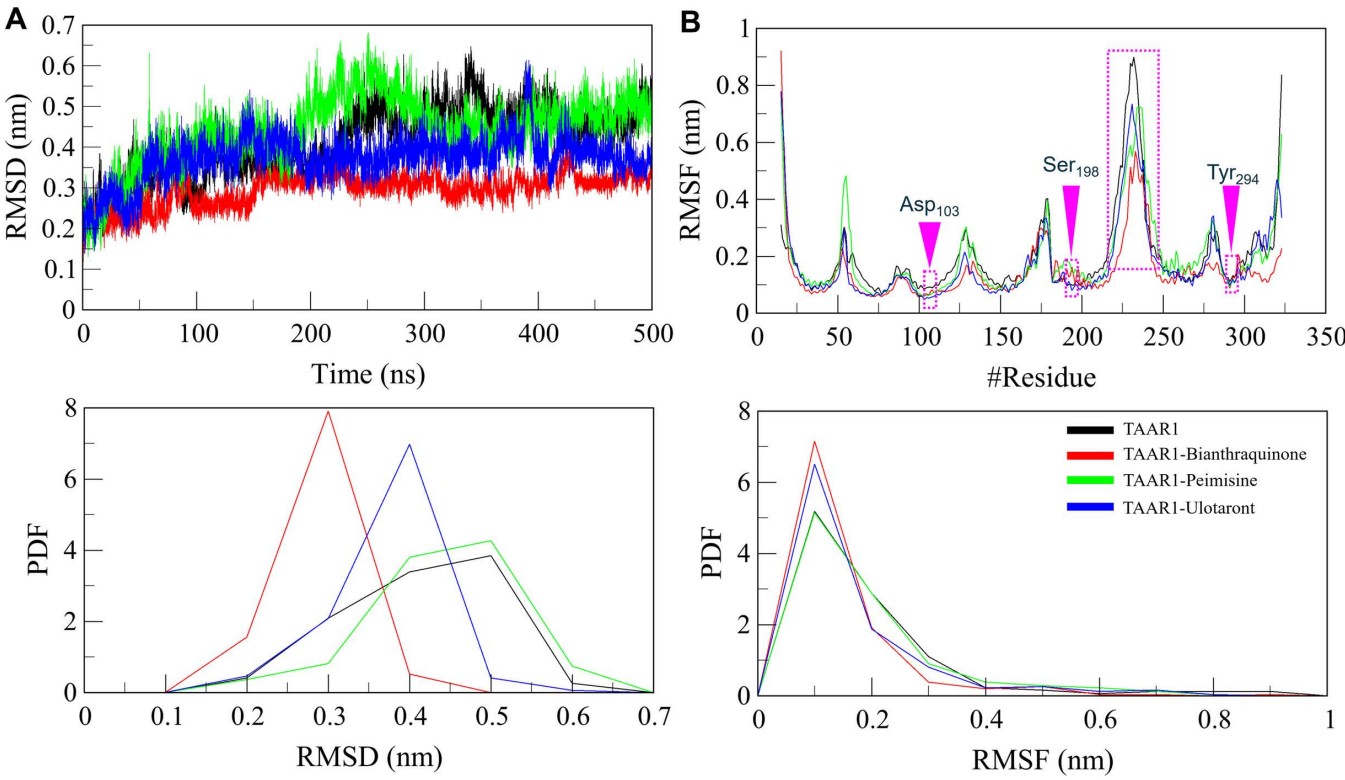

**Fig 3. Structural dynamics analysis of TAAR1 upon ligand binding.** (A) RMSD plot comparing TAAR1 stability in its free state and complex with Bianthraquinone, Peimisine, and Ulotaront over a 500 ns MD simulation. (B) RMSF analysis of TAAR1 shows specific residues' flexibility before and after ligand binding. The ligand binding site residues in the RMSF are highlighted in magenta dotted boxes. The lower panels show the probability density function (PDF) distribution of RMSD and RMSF values.

of $R_g$ of values (Fig 4A, lower panel) showed similar distributions before and after ligand binding (indicating that Bianthraquinone and Peimisine did not lead to global architecture changes of TAAR1). At the confirmation of ligand-binding, these identified compounds were responsible for stabilizing and retaining a well-folded state of the proteins, i.e., these phytochemicals did not load the proteins with large-scale conformational perturbations, which might prevent them from exerting the biological effect.

SASA analysis was performed as an indicator of protein stability and to evaluate the extent of TAAR1 surface exposure to the solvent throughout the simulation. SASA is a critical parameter influencing protein folding; limited solvent exposure generally indicates greater structural stability [51]. Accordingly, lower overall SASA values suggest more compact folding and enhanced protein stability. We compared the SASA profiles of TAAR1, TAAR1-Bianthraquinone, TAAR1-Peimisine, and TAAR1-Ulotaront and found that the ligand binding did not cause significant perturbations of the protein's solvent exposure (Fig 4B). A similar distribution gradient of SASA values was also observed before and after ligand binding, confirming that TAAR1 retained structural stability in both free and ligand-attached conformation (Fig 4B, lower panel). These findings suggest that TAAR1 remains compact and maintains structural integrity upon binding Bianthraquinone and Peimisine, with no substantial local or global conformational changes indicating instability. The stable $R_g$ and SASA values also confirm the stabilization effect of the selected ligand molecules, thus suggesting them as potential TAAR1-targeting candidates. As these compounds can maintain the structural integrity of TAAR1, they open the door for experimental validation and optimization in the drug development process.

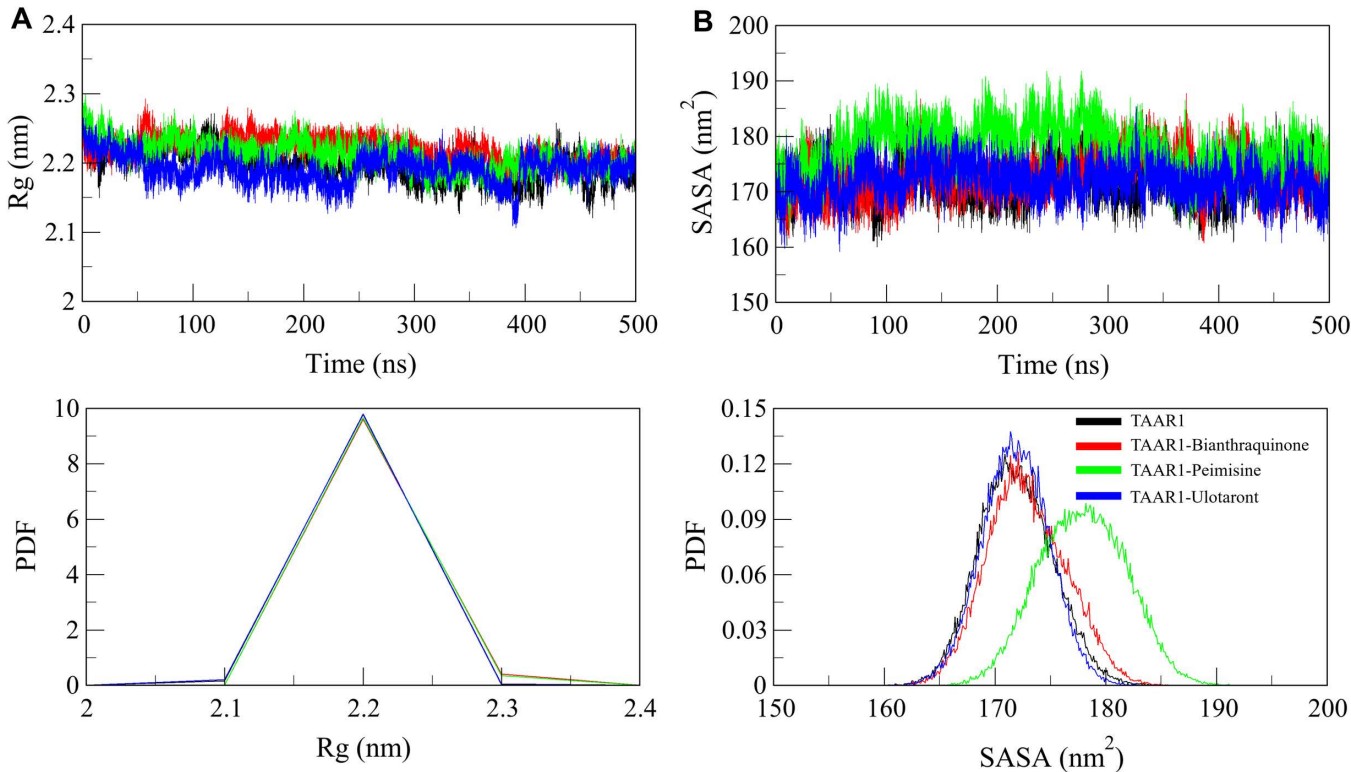

**Fig 4. Evaluation of TAAR1's structural compactness before and after ligand binding.** (A) $R_g$ plot showing the compactness of TAAR1 in the presence and absence of Bianthraquinone, Peimisine, and Ulotaront. (B) SASA analysis illustrating the extent of protein exposure to the solvent over time. The lower panels show the probability density function (PDF) distribution of $R_g$ and SASA values.

**Table 5. Average values for different MD parameters of TAAR1 and TAAR1-ligand complexes.**

| S. No. | Protein/protein-ligand system | RMSD (nm) | RMSF (nm) | $Rg$ (nm) | SASA (nm²) | #Intramolecular H-bonds |
|---|---|---|---|---|---|---|
| 1. | TAAR1 | 0.41±0.09 | 0.20±0.15 | 2.20±0.02 | 171.89±3.31 | 235±11 |
| 2. | TAAR1-Bianthraquinone | 0.30±0.04 | 0.15±0.10 | 2.22±0.02 | 172.85±3.57 | 247±7 |
| 3. | TAAR1-Peimisine | 0.44±0.08 | 0.19±0.13 | 2.21±0.02 | 177.91±3.92 | 239±8 |
| 4. | TAAR1-Ulotaront | 0.37±0.06 | 0.17±0.13 | 2.19±0.02 | 171.86±3.01 | 246±7 |

### 3.5.3. Hydrogen bonding.

Hydrogen-bonding is vital in preserving protein structural stability and receptor-ligand interactions [52]. Hydrogen bonds (H-bonds) occurring between protein chains (intramolecular) and between the ligand and the protein (intermolecular) play a pivotal role in driving protein folding, conformational stability, and ligand binding [53]. The intramolecular H-bonding profile of TAAR1 was analyzed without and in the presence of ligand to ascertain whether Bianthraquinone and Peimisine influenced the stability of the protein (Fig 5). The data showed that TAAR1-Bianthraquinone and TAAR1-Peimisine complexes possessed more intramolecular H-bonds than unbound TAAR1 (Fig 5A–B). This indicates that ligand binding was associated with enhanced compactness and rigidity of the receptor. The higher H-bond network stability is usually associated with a well-structured and thermodynamically stable protein structure, substantiating the stabilizing effects of the predicted phytochemicals.

Besides intramolecular interaction, intermolecular H-bonds between TAAR1 and the bound ligands were investigated to assess the robustness and longevity of ligand interactions. The H-bond occupancy graph showed the formation of stable H-bonds between Bianthraquinone and TAAR1, as well as between Peimisine and TAAR1, in a long-range trajectory

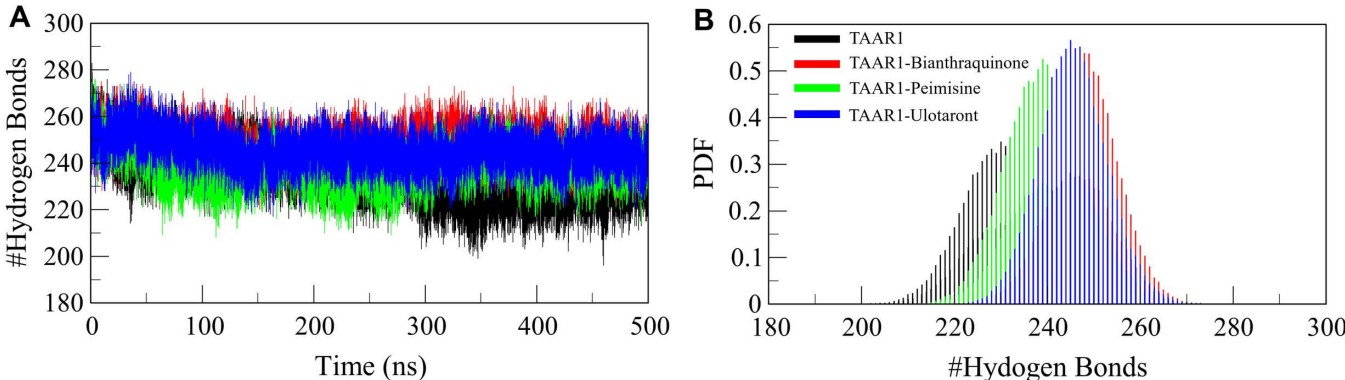

**Fig 5. Hydrogen bonding dynamics between TAAR1 and its ligand-bound complexes.** (A) Time evolution of intramolecular H-bonds in TAAR1 before and after binding with Bianthraquinone, Peimisine, and Ulotaront. (B) The panel showed the probability density function (PDF) distribution of intramolecular H-bonds in TAAR1.

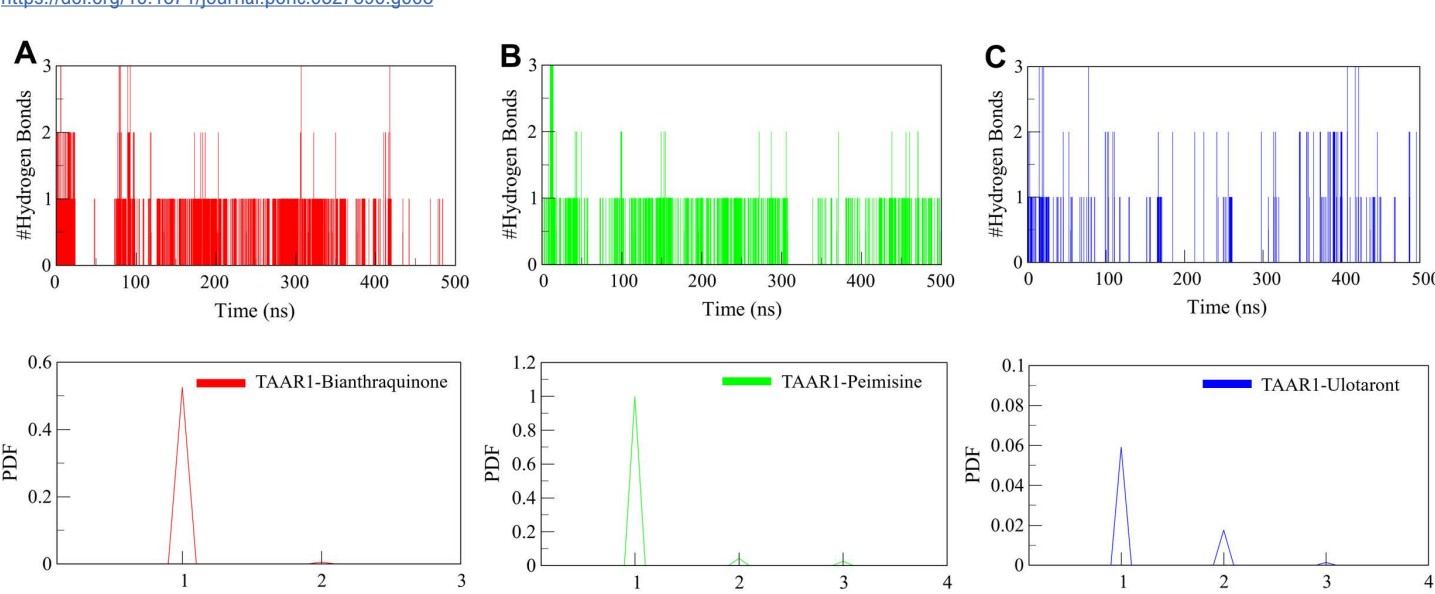

**Fig 6. Hydrogen bonding dynamics between TAAR1 and its ligands.** (A) Time-evolution of intermolecular H-bonds between TAAR1 and Bianthraquinone. (B) Time-evolution of intermolecular H-bonds between TAAR1 and Peimisine. (C) Time-evolution of intermolecular H-bonds between TAAR1 and Ulotaront. The lower panels illustrate the probability distribution of intermolecular H-bonds.

of these imprinted conformations, suggesting a strong binding affinity (Fig 6). The' consistency and sustaining nature of ligand-receptor interactions over the simulation period is further confirmed with the PDF of H-bond counts (Fig 6, lower panels). Intermolecular H-bond stability is a crucial factor for ligand binding free energy. The stable hydrogen bonds formed in both TAAR1-Bianthraquinone and TAAR1-Peimisine complexes can enhance the conformational stability of the complexes, thus decreasing receptor conformational mobility and stabilizing the interaction network between the receptor domains. The higher H-bond occupancy and reduced flexibility of ligand-bound TAAR1 suggest that Bianthraquinone and Peimisine could serve as superior TAAR1 agonists in future drug development. In summary, the intermolecular interactions with Bianthraquinone and Peimisine concluded the enhanced structural stability and binding efficiency of phytochemicals, thus enabling us to call them promising contenders for further evaluation.

## 3.6. Principal component analysis

PCA was applied to the MD trajectories to identify the dominant modes of motion within the conformational space of the biomolecular system. PCA enables the extraction of the most significant collective motions, thereby aiding in understanding structural plasticity, ligand-induced conformational changes, and their potential functional implications [40]. PCA was performed in this study to examine the conformational space sampled by TAAR1 when unbound and when bound to the Bianthraquinone and Peimisine ligands (Fig 7). These molecular motions were predicted from the Cα atomic fluctuations to observe how the motion was flexible and stable in the unbound or ligand state. Fig 7A shows the conformational sampling of TAAR1 and its complexes in the essential subspace, as well as unbound TAAR1 that sampled a greater conformational space, as expected due to increased flexibility and structural fluctuations. In comparison, the TAAR1-Bianthraquinone complex populated a much tighter conformational region than TAAR1-Peimisine and TAAR1-Ulotaront, implying a ligand-binding correlative reduction of flexibility.

The overlapping of the TAAR1-Peimisine complex with the free TAAR1 subspace suggested that little conformational change occurred upon Peimisine binding. In contrast, the TAAR1-Bianthraquinone complex adopted a markedly more compact, distinct pathway along principal component 1 (EV1), consistent with a strong stabilization effect (Fig 7B). Ligand binding-induced diminishing of flexibility indicates stabilizing contribution of Bianthraquinone and Peimisine to TAAR1's conformational states, with more significant impact from Bianthraquinone. We found that the ligand step can restrict structural motions on a large scale, which is usually associated with increased binding stability and therapeutic efficacy. These results reinforce that Bianthraquinone and Peimisine act as potent TAAR1 stabilizers and lead compounds for further optimization.

## 3.7. Free energy landscapes

FEL is a vital thermodynamic tool, depicting a molecular system's thermodynamic stability and energetics as a function of a few essential collective coordinates within the conformational space [40]. FELS can be analysed to determine a protein-ligand complex's most stable structural states and how binding a ligand alters its conformational equilibrium. For analysis of

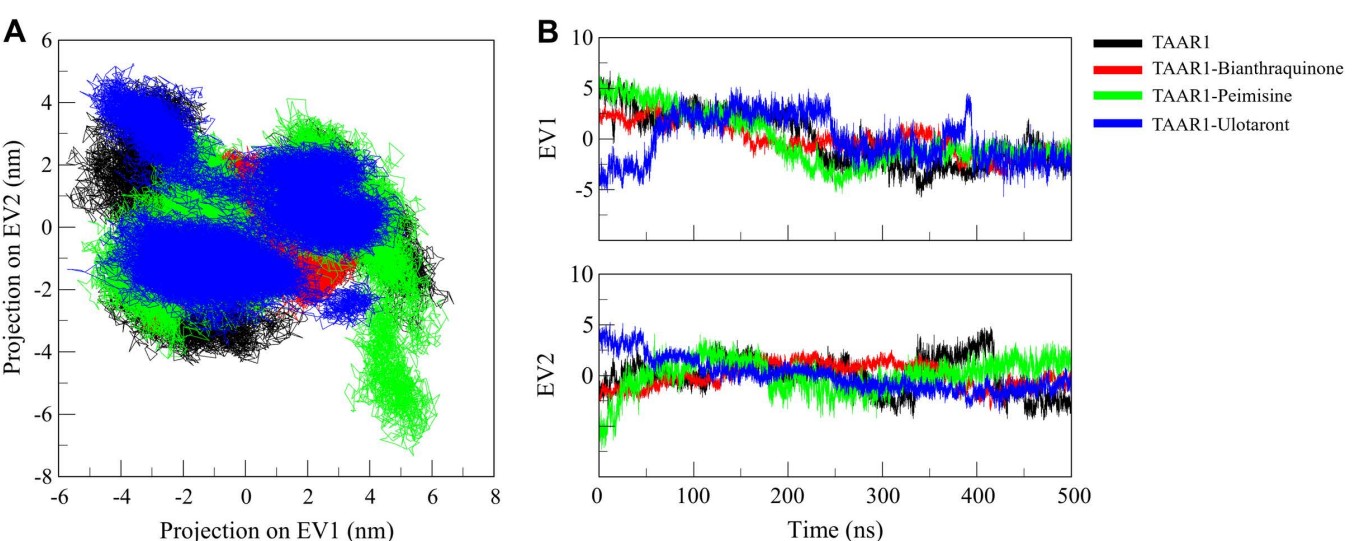

**Fig 7. Conformational sampling of TAAR1 and its ligand-bound complexes in PCA.** (A) 2D projection of TAAR1, TAAR1-Bianthraquinone, TAAR1-Peimisine and TAAR1-Ulotaront trajectories in the essential conformational space. (B) Time-evolution of PCA of TAAR1 and its ligand-bound complexes.

the energetic stability of the TAAR1, TAAR1-Bianthraquinone, TAAR1-Peimisine, and TAAR1-Ulotaront systems, FELs were constructed from the MD simulation trajectories, using the top two principal components (PC1 and PC2) as reaction coordinates (Fig 8). These energy minima correspond to the least thermodynamically unstable conformations. These results suggested higher flexibility and the existence of various conformational states of TAAR1 manifested by several spread energy minima (Fig 8A). Upon ligand binding, TAAR1-Bianthraquinone and TAAR1-Peimisine complexes revealed a more constrained energy landscape with one or two prominent minima, indicating enhanced structural stability (Fig 8B–C).

Among the global minima of the TAAR1-ligand complexes, the TAAR1-Bianthraquinone complex featured the most compact and well-defined structure, suggesting that TAAR1-Bianthraquinone interacts with more excellent binding stability than Peimisine. The structure of the TAAR1 confirms that Bianthraquinone and Peimisine binding limit the conformational liberty of TAAR1, resulting in a more stable structural ensemble in the heteromeric. In TAAR1-Bianthraquinone, the stabilisation effect was even more pronounced, strengthening the evidence for TAAR1-Bianthraquinone as a better TAAR1 agonist than Peimisine and the reference compound Ulotaront (Fig 8D). PCA results indicate an encouraging promise for Bianthraquinone as a TAAR1-targeting phytochemical with no or substantially lowered energetic deviation and majorly reduced conformational alterations. More detailed quantitative calculations on ligand-binding energies using free energy perturbation (FEP) [54] can be performed in addition to experimental validation to corroborate these findings.

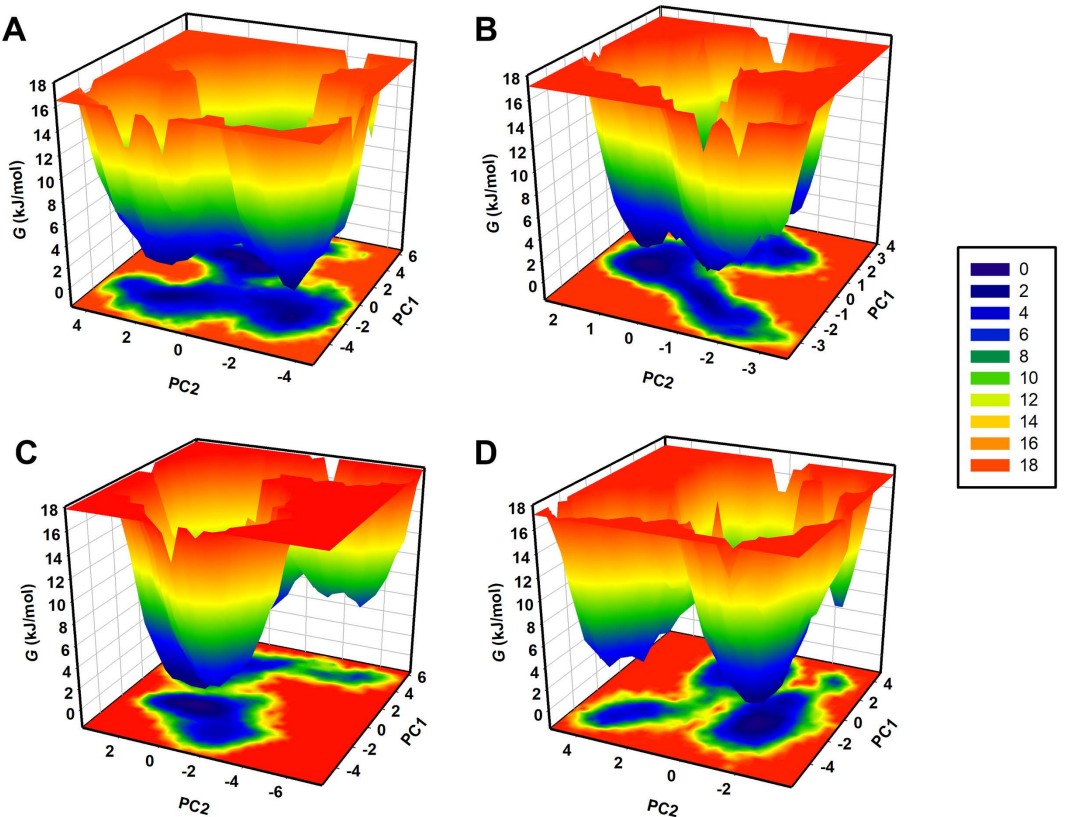

**Fig 8. Free energy landscapes of TAAR1.** (A) FEL representation of TAAR1, indicating multiple energy minima. (B) The FEL representation of TAAR1 in complex with Bianthraquinone shows a restricted energy landscape with a well-defined global minimum. (C) FEL representation of TAAR1 in complex with Peimisine. (D) FEL representation of TAAR1 in complex with Ulotaront.

## 3.8. MM-PBSA analysis

The binding free energy ($\Delta G_{binding}$) is a key thermodynamic indicator of ligand–receptor interactions favorability, and lower values denote stronger and more stable binding [41]. This approach decomposes total binding energy into van der Waals interaction, electrostatic energy, polar solvation energy, and SASA energy. All these individual components collectively play a role in determining the overall stability of the complex. MM-PBSA calculations were performed for the TAAR1-Bianthraquinone, TAAR1-Peimisine, and TAAR1-Ulotaront complexes to get insights into the binding stability and inter-action energetics. As is summarized in Table 6, all three ligand-TAAR1 complexes displayed negative, or energetically favorable, binding free energies. The extent of binding affinity and stability was quantified by more negative binding energies than TAAR1-Bianthraquinone among TAAR1-Peimisine and TAAR1-Ulotaront. Notably, Bianthraquinone formed more stable complexes than TAAR1-Peimisine and TAAR1-Ulotaront and hence may be a more potent TAAR1 agonist. It also suggests that the dominant favorable forces in binding were van der Waals and electrostatic, and that polar solvation energies help to some extent in opposing binding. However, these components were nonetheless not enough to over-come the net binding energies, which remained strongly negative, as a result of the balance between these components, indicating that the ligands could form stable and specific interactions with TAAR1.

## 4. Conclusions

This study highlights the potential of phytochemicals as novel therapeutic candidates for MDD through their interaction with TAAR1, a promising nondopaminergic target. Utilizing a structure-guided virtual screening approach, we identified two lead compounds, Bianthraquinone and Peimisine, from the IMPPAT database that demonstrated strong binding affinity, favorable pharmacokinetic properties, and high predicted biological activity compared to the reference agonist, Ulotaront. These compounds exhibited high binding affinities, good pharmacokinetic profiles, and stability in MD simula-tions, suggesting they are promising drug candidates. Subsequent essential dynamics and MM-PBSA analyses confirmed the binding stability and energetic favorability of these phytochemicals and conformational compatibility with TAAR1. Bian-thraquinone and Peimisine displayed strong receptor interactions, favorable pharmacokinetics, and significant stabilization of TAAR1 structure. These findings provide a compelling basis for their development as novel TAAR1-targeting antide-pressants. Future directions will involve *in vitro* validation, chemical optimization, and *in vivo* efficacy studies.

## Supporting information

**S1 Fig. TAAR1 in complex with co-crystallized (magenta) and docked (cyan) Ulotaront.** The figure was generated in PyMOL using the PDB structure with ID: 8JLO. The co-crystallized and docked Ulotaront molecules are superimposed, showing a root-mean-square deviation (RMSD) of 0.198 Å.
(DOCX)

## Acknowledgments

The authors extend their appreciation to the King Salman center For Disability Research for funding this work through Research Group no KSRG-2024–285. AS thanks to the Ajman University for APC.

**Table 6. MM-PBSA parameters calculated for the TAAR1 ligand-bound complexes.**

| Complex | $\Delta G_{VDWAALS}$ | $\Delta E_{EL}$ | $\Delta E_{PB}$ | $\Delta E_{NPOLAR}$ | $\Delta G_{GAS}$ | $\Delta G_{SOLV}$ | $\Delta G_{Total}$ (kJ/mol) |
|---|---|---|---|---|---|---|---|
| TAAR1-Bianthraquinone | −31.26 | −11.33 | 23.49 | −3.49 | −42.59 | 20.01 | −22.58±3.30 |
| TAAR1-Peimisine | −10.50 | −3.16 | 8.32 | −1.35 | −13.66 | 6.97 | −6.69±3.91 |
| TAAR1-Ulotaront | −22.55 | 1.67 | 7.08 | −2.33 | −20.88 | 4.75 | −16.13±2.04 |

## Author contributions

**Conceptualization:** Abdelbaset Mohamed Elasbali, Mohd Adnan, Md. Imtaiyaz Hassan.

**Data curation:** Abdelbaset Mohamed Elasbali, Ahmed S. Ali, Taj Mohammad, Md. Imtaiyaz Hassan.

**Formal analysis:** Ahmed S. Ali, Anas Shamsi, Md. Imtaiyaz Hassan.

**Funding acquisition:** Abdelbaset Mohamed Elasbali, Taj Mohammad, Anas Shamsi.

**Investigation:** Abdelbaset Mohamed Elasbali, Mohd Adnan, Taj Mohammad.

**Methodology:** Ahmed S. Ali, Mohd Adnan, Taj Mohammad, Md. Imtaiyaz Hassan.

**Resources:** Mohd Adnan, Md. Imtaiyaz Hassan.

**Software:** Ahmed S. Ali.

**Supervision:** Anas Shamsi.

**Validation:** Taj Mohammad, Anas Shamsi, Md. Imtaiyaz Hassan.

**Visualization:** Abdelbaset Mohamed Elasbali, Mohd Adnan, Anas Shamsi, Md. Imtaiyaz Hassan.

**Writing – original draft:** Abdelbaset Mohamed Elasbali, Ahmed S. Ali, Taj Mohammad.

**Writing – review & editing:** Mohd Adnan, Anas Shamsi, Md. Imtaiyaz Hassan.

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
