## [Decision Letter · Decision Letter 0]

29 May 2025

Dear Dr. Hassan,

Thank you for submitting your manuscript to PLOS ONE. After careful consideration, we feel that it has merit but does not fully meet PLOS ONE’s publication criteria as it currently stands. Therefore, we invite you to submit a revised version of the manuscript that addresses the points raised during the review process.

We look forward to receiving your revised manuscript.

Kind regards,

Ahmed A. Al-Karmalawy, PhD

Academic Editor

PLOS ONE

**Journal requirements:**

1. When submitting your revision, we need you to address these additional requirements. Please ensure that your manuscript meets PLOS ONE's style requirements, including those for file naming. The PLOS ONE style templates can be found at https://journals.plos.org/plosone/s/file?id=wjVg/PLOSOne_formatting_sample_main_body.pdf and https://journals.plos.org/plosone/s/file?id=ba62/PLOSOne_formatting_sample_title_authors_affiliations.pdf 2. Please note that PLOS ONE has specific guidelines on code sharing for submissions in which author-generated code underpins the findings in the manuscript. In these cases, we expect all author-generated code to be made available without restrictions upon publication of the work. Please review our guidelines at https://journals.plos.org/plosone/s/materials-and-software-sharing#loc-sharing-code and ensure that your code is shared in a way that follows best practice and facilitates reproducibility and reuse. 3. We note that the grant information you provided in the ‘Funding Information’ and ‘Financial Disclosure’ sections do not match.  When you resubmit, please ensure that you provide the correct grant numbers for the awards you received for your study in the ‘Funding Information’ section. 4. Thank you for stating the following in the Acknowledgments Section of your manuscript: The authors extend their appreciation to the King Salman center For Disability Research for funding this work through Research Group no KSRG-2024-285. We note that you have provided funding information that is not currently declared in your Funding Statement. However, funding information should not appear in the Acknowledgments section or other areas of your manuscript. We will only publish funding information present in the Funding Statement section of the online submission form. Please remove any funding-related text from the manuscript and let us know how you would like to update your Funding Statement. Currently, your Funding Statement reads as follows: The author(s) received no specific funding for this work.  Please include your amended statements within your cover letter; we will change the online submission form on your behalf. 5. We note that your Data Availability Statement is currently as follows: All relevant data are within the manuscript and its Supporting Information files. Please confirm at this time whether or not your submission contains all raw data required to replicate the results of your study. Authors must share the “minimal data set” for their submission. PLOS defines the minimal data set to consist of the data required to replicate all study findings reported in the article, as well as related metadata and methods (https://journals.plos.org/plosone/s/data-availability#loc-minimal-data-set-definition). For example, authors should submit the following data: - The values behind the means, standard deviations and other measures reported;- The values used to build graphs;- The points extracted from images for analysis. Authors do not need to submit their entire data set if only a portion of the data was used in the reported study. If your submission does not contain these data, please either upload them as Supporting Information files or deposit them to a stable, public repository and provide us with the relevant URLs, DOIs, or accession numbers. For a list of recommended repositories, please see https://journals.plos.org/plosone/s/recommended-repositories. If there are ethical or legal restrictions on sharing a de-identified data set, please explain them in detail (e.g., data contain potentially sensitive information, data are owned by a third-party organization, etc.) and who has imposed them (e.g., an ethics committee). Please also provide contact information for a data access committee, ethics committee, or other institutional body to which data requests may be sent. If data are owned by a third party, please indicate how others may request data access.

Reviewers' comments:

Reviewer's Responses to Questions

**Comments to the Author**

1. Is the manuscript technically sound, and do the data support the conclusions?

Reviewer #1: Partly

Reviewer #2: Yes

2. Has the statistical analysis been performed appropriately and rigorously?

Reviewer #1: Yes

Reviewer #2: Yes

3. Have the authors made all data underlying the findings in their manuscript fully available?

Reviewer #1: Yes

Reviewer #2: Yes

4. Is the manuscript presented in an intelligible fashion and written in standard English?

Reviewer #1: Yes

Reviewer #2: Yes

**Reviewer #1:**  The manuscript presents a robust computational pipeline for identifying TAAR1 agonists, but addressing the above points will strengthen its rigor and translational potential. Emphasizing the need for experimental follow-up and discussing limitations will enhance the study’s credibility:

1. Mention the method that used to evaluate compounds through Lipinski's rule (please mention it in methodology)

2. How to predict the binding site of TAAR1 (please , mention crucial amino acid residues that responsible for binding)

3. Why was InstaDock chosen over established tools like AutoDock or Glide for molecular docking? How was its reliability validated (e.g., through redocking of Ulotaront)

4. Could you elaborate on the structural features critical for TAAR1 binding? Are there opportunities for optimizing these scaffolds? (write discussion important SAR for TAAR1 binding)

**Reviewer #2: ** To the Editor: Dear Dr. Hassan and Collaborators,

I trust this email reaches you well.

Having critically assessed your manuscript entitled “Structure-Based Identification of Bioactive Compounds as Trace Amine-Associated Receptor 1 Agonists for the Therapeutic Management of Major Depressive Disorder” (Manuscript ID: PONE-D-25-21674), and evaluated the reviewers’ comments, I would like to summarize the major points to be revised, including some suggestions to be considered for improvement. These edits serve to increase the clarity, scientific accuracy, and the overall quality of the manuscript prior to submission.General Observations

The manuscript presents promising findings on phytochemical agonists targeting TAAR1 and their relevance to MDD.

The overall methodology is sound and comprehensive.

Improvements are needed in grammar, clarity, redundancy elimination, and data presentation for better scientific communication.

Summary of Required Revisions by Section

1. Abstract

Correct grammatical issues and ensure complete sentence structure (e.g., Line 34–35).

Refine conclusion sentence for impact.

2. Introduction

Remove unnecessary sentence structure and over-long sentences (e.g., Line 66, 89).

Clarify the role of TAAR1 in neuropsychiatry and why natural products are promising.

3. Materials and Methods

Define resolution and TAAR1 structure details (Line 117).

State docking box size and list all used software with URLs.

Mentioning known PRODRG limitations and PCA/FEL assumptions might be useful.

4. Results and Discussion

Quantitatively report data for RMSD, H-bonding, Rg, and SASA with SD as appropriate.

Breaking up complex interaction data into tables rather than wordy paragraphs may be a good idea.

5. Conclusions

Correct typing errors and redundancy (e.g., "Peimisine and Peimisine").

Close with greater emphasis on translational potential and future work.

6. Declarations

Declare consistent with "Funding: NONE" and the Acknowledgment thanking the King Salman Center.

I recommend that your manuscript be accepted

best regards

**Do you want your identity to be public for this peer review?** For information about this choice, including consent withdrawal, please see our Privacy Policy

Reviewer #1: No

Reviewer #2: **Yes**

---

## [Author Response · Author response to Decision Letter 1]

2 Jun 2025

Reviewer #1:

The manuscript presents a robust computational pipeline for identifying TAAR1 agonists, but addressing the above points will strengthen its rigor and translational potential. Emphasizing the need for experimental follow-up and discussing limitations will enhance the study’s credibility:

1. Mention the method that used to evaluate compounds through Lipinski's rule (please mention it in methodology)

Response: Thank you for your valuable suggestion. As advised, we have mentioned the use of Lipinski’s rule of five for the initial filtering of phytochemicals in the methodology section of the revised manuscript.

2. How to predict the binding site of TAAR1 (please, mention crucial amino acid residues that responsible for binding)

Response: We appreciate this insightful comment. The binding of TAAR1 is known as annotated at UniProt (https://www.uniprot.org/uniprotkb/Q96RJ0) based on multiple sources (PMID: 37935376, PMID: 37963465, PMID: 37935377). The manuscript has now been updated to clearly describe the key binding site residues of TAAR1, especially focusing on Asp103, Ser107, Tyr294, Trp264, Phe268, and others involved in hydrogen bonding and hydrophobic contacts. This information has been incorporated in the revised manuscript.

3. Why was InstaDock chosen over established tools like AutoDock or Glide for molecular docking? How was its reliability validated (e.g., through redocking of Ulotaront)

Response: Thank you for pointing this out. We have updated the Molecular Docking section (2.2) to justify our use of InstaDock, which offers an efficient and user-friendly graphical interface suitable for high-throughput screening. Its core algorithm is built on AutoDock Vina, ensuring comparable reliability. We have also performed the redocking analysis to validate the docking parameters and binding site selection, as recommended. The predicted binding poses are closely aligned with the experimentally determined crystallographic conformation of Ulotaront. The results have been mentioned in section 2.2 and Supplementary Fig. S1.

4. Could you elaborate on the structural features critical for TAAR1 binding? Are there opportunities for optimizing these scaffolds? (write discussion important SAR for TAAR1 binding)

Response: We thank the reviewer for highlighting this important point. Now, we have included a dedicated discussion within Section 3.4 on the SAR relevant to TAAR1 binding. We have emphasized the recurring presence of hydrophobic aromatic rings, H-bond donors near Asp103, and pi-pi stacking motifs, which appear critical for receptor activation. We have also discussed potential optimization strategies by modifying functional groups to enhance affinity and selectivity. Thanks!

Reviewer #2:

Having critically assessed your manuscript entitled “Structure-Based Identification of Bioactive Compounds as Trace Amine-Associated Receptor 1 Agonists for the Therapeutic Management of Major Depressive Disorder” (Manuscript ID: PONE-D-25-21674), and evaluated the reviewers’ comments, I would like to summarize the major points to be revised, including some suggestions to be considered for improvement. These edits serve to increase the clarity, scientific accuracy, and the overall quality of the manuscript prior to submission.

General Observations

The manuscript presents promising findings on phytochemical agonists targeting TAAR1 and their relevance to MDD.

The overall methodology is sound and comprehensive.

Improvements are needed in grammar, clarity, redundancy elimination, and data presentation for better scientific communication.

Response: We sincerely thank the reviewer for their overall positive assessment and valuable feedback. The manuscript has undergone a thorough grammatical revision, removal of redundancies, and clarification of ambiguous statements across all sections. Additionally, we have improved data presentation by breaking down complex descriptions into well-organized tables and figures during this revision.

Summary of Required Revisions by Section

1. Abstract

Correct grammatical issues and ensure complete sentence structure (e.g., Line 34–35).

Refine conclusion sentence for impact.

Response: We appreciate this suggestion. The Abstract has been revised for improved grammar, sentence clarity, and overall flow. The concluding sentence now emphasizes the translational significance and therapeutic promise of the identified phytochemicals as next-generation TAAR1 agonists.

2. Introduction

Remove unnecessary sentence structure and over-long sentences (e.g., Line 66, 89).

Clarify the role of TAAR1 in neuropsychiatry and why natural products are promising.

Response: We have carefully revised the Introduction to simplify and shorten complex sentences. The role of TAAR1 in neuropsychiatry has been articulated more clearly, emphasizing its advantages over monoaminergic targets. Furthermore, we highlighted the rationale for focusing on natural products, citing their historical success and pharmacological richness.

3. Materials and Methods

Define resolution and TAAR1 structure details (Line 117).

State docking box size and list all used software with URLs.

Mentioning known PRODRG limitations and PCA/FEL assumptions might be useful.

Response: Thank you for this comprehensive comment. We have made the following additions:

• Resolution and details of TAAR1 (PDB: 9JKQ) are now mentioned in Section 2.1.

• Docking box size and coordinates are described in Section 2.2.

• All tools and software with URLs have been listed where they are first mentioned.

• The known limitations of PRODRG and the assumptions made in PCA/FEL have been briefly discussed.

4. Results and Discussion

Quantitatively report data for RMSD, H-bonding, Rg, and SASA with SD as appropriate.

Breaking up complex interaction data into tables rather than wordy paragraphs may be a good idea.

Response: We are grateful for this practical suggestion. Accordingly, we have added quantitative results for RMSD, RMSF, Rg, SASA, and hydrogen bonding in Table 5 of the revised manuscript. Also, the protein-ligand interaction data have been summarized in Table 4 of the revised manuscript.

5. Conclusions

Correct typing errors and redundancy (e.g., "Peimisine and Peimisine").

Close with greater emphasis on translational potential and future work.

Response: Thank you for catching this typo. We have corrected the repetition and revised the Conclusion section to better highlight the translational potential of Bianthraquinone and Peimisine as TAAR1 agonists. We also added a note on future experimental validation and optimization of these scaffolds as part of a pipeline for antidepressant development.

6. Declarations

Declare consistent with "Funding: NONE" and the Acknowledgment thanking the King Salman Center.

I recommend that your manuscript be accepted

best regards

Response: We appreciate this observation. The funding statement has been updated to reflect accurate details. This is now consistently declared in both the Funding and Acknowledgments sections. Thanks!

---

## [Decision Letter · Decision Letter 1]

24 Jun 2025

Structure-Based Identification of Bioactive Compounds as Trace Amine-Associated Receptor 1 Agonists for the Therapeutic Management of Major Depressive Disorder

PONE-D-25-21674R1

Dear Dr. Hassan,

We’re pleased to inform you that your manuscript has been judged scientifically suitable for publication and will be formally accepted for publication once it meets all outstanding technical requirements.

Kind regards,

Ahmed A. Al-Karmalawy, PhD

Academic Editor

PLOS ONE

Reviewers' comments:

Reviewer's Responses to Questions

**Comments to the Author**

Reviewer #1: All comments have been addressed

Reviewer #2: All comments have been addressed

2. Is the manuscript technically sound, and do the data support the conclusions?

Reviewer #1: Yes

Reviewer #2: Yes

3. Has the statistical analysis been performed appropriately and rigorously?

Reviewer #1: Yes

Reviewer #2: Yes

4. Have the authors made all data underlying the findings in their manuscript fully available?

Reviewer #1: Yes

Reviewer #2: Yes

5. Is the manuscript presented in an intelligible fashion and written in standard English?

Reviewer #1: Yes

Reviewer #2: Yes

Reviewer #1: (No Response)

Reviewer #2: (No Response)

**Do you want your identity to be public for this peer review?** For information about this choice, including consent withdrawal, please see our Privacy Policy

Reviewer #1: **Yes**

Reviewer #2: No

---

## [Editor Report · Acceptance letter]

PONE-D-25-21674R1

PLOS ONE

Dear Dr. Hassan,

I'm pleased to inform you that your manuscript has been deemed suitable for publication in PLOS ONE. Congratulations! Your manuscript is now being handed over to our production team.

Kind regards,

on behalf of

Associate Professor Ahmed A. Al-Karmalawy

Academic Editor

PLOS ONE